# A Local Planner for Accurate Positioning for a Multiple Steer-and-Drive Unit Vehicle Using Non-Linear Optimization

**DOI:** 10.3390/s22072588

**Published:** 2022-03-28

**Authors:** Henrik Andreasson, Jonas Larsson, Stephanie Lowry

**Affiliations:** 1Centre for Applied Autonomous Sensor Systems (AASS), Örebro University, 701 82 Örebro, Sweden; stephanie.lowry@oru.se; 2ABB Corporate Research, 722 26 Västerås, Sweden; jonas.larsson@se.abb.com

**Keywords:** local planning, optimal control, obstacle avoidance

## Abstract

This paper presents a local planning approach that is targeted for pseudo-omnidirectional vehicles: that is, vehicles that can drive sideways and rotate on the spot. This local planner—MSDU–is based on optimal control and formulates a non-linear optimization problem formulation that exploits the omni-motion capabilities of the vehicle to drive the vehicle to the goal in a smooth and efficient manner while avoiding obstacles and singularities. MSDU is designed for a real platform for mobile manipulation where one key function is the capability to drive in narrow and confined areas. The real-world evaluations show that MSDU planned paths that were smoother and more accurate than a comparable local path planner Timed Elastic Band (TEB), with a mean (translational, angular) error for MSDU of (0.0028 m, 0.0010 rad) compared to (0.0033 m, 0.0038 rad) for TEB. MSDU also generated paths that were consistently shorter than TEB, with a mean (translational, angular) distance traveled of (0.6026 m, 1.6130 rad) for MSDU compared to (0.7346 m, 3.7598 rad) for TEB.

## 1. Introduction

Omni-motion capability—the ability to drive in all directions, not just forwards—allows greater maneuverability for mobile robots and similar platforms [1,2]. A platform with restricted omni-motion capability must make more complex maneuvers to achieve goal positions [1], which can be difficult to plan [3], time-consuming, and even impossible in crowded or constrained physical environments [3]. Furthermore, greater mobile maneuverability allows a robot to achieve more accurate pose targets [4] which are essential for many industrial applications.

A fully holonomic platform has maximal omni-motion capability as such a platform can move freely in any direction [2], and fully holonomic motion can be achieved using omnidirectional wheels such as the Mecanum (or Swedish) wheel [5]. However, there are drawbacks to such a wheel configuration: these wheels are mechanically complex and the wheels are sensitive to non-smooth surfaces and thus require either clean, flat floors or complex suspension systems that ensure that the wheels are in contact with the ground [6]. For this reason, while fully holonomic systems are common in academic research [7], very few commercial platforms exist, although an exception is the Omnibot from KUKA [8].

Alternatively to a fully holonomic platform, a platform with a wheel configuration that uses multiple combined steer and drive wheels is an attractive option for an omni-motion robotic platform. It is not fully holonomic, but it has many advantages: it is robust to floor and environment conditions; it has good motion performance, with high acceleration, deacceleration, and turning capability; and it provides accurate wheel odometry, compared to the slip experienced by fully holonomic wheels.

However, a challenge with steerable wheels is the increased complexity in vehicle motion control. This is due to the resulting non-holonomic nature of the vehicle motion as well as the inverse kinematics equations possessing singularities. Therefore, specialized motion-planning algorithms are required to ensure that the robot motion planning and control optimizes the agility and other advantages of the combined steer and drive wheels, while ensuring that the planner produces feasible paths for the platform to follow, and avoids singularities.

Existing local motion-planning algorithms often provide generic paths that do not fully exploit the pseudo-omnidirectional capabilities of the platform [9] or the kinodynamic constraints [10]. Furthermore, many local motion-planning algorithms do not inherently handle obstacles [11], which can be critical when navigating in confined and narrow environments. Furthermore, local motion planners often try to solve the global planning problem, which is computationally inefficient or infeasible for long paths [10]. Alternatively, local motion planners which do not solve the global path planning problem can cause the platform to get stuck in local minima and fail to reach the goal position [12,13].

The contribution of this paper is a local motion planner that exploits the pseudo-omnidirectional capabilities of the platform to generate an efficient trajectory, allow precise positioning, and avoid singular configurations. It combines constraints from both vehicle dynamics and obstacles observed by the robot’s perception module to ensure that the outputted trajectories are kinodynamically feasible as well as safe from obstacle collisions.

The local motion planner is designed to integrate with a global motion planner that provides an initial path estimate based on the robot’s internal map of the environment. Coupling with a global planner avoids the need to address longer, time-consuming trajectory optimization steps or to use additional reasoning in the framework to select a shorter path [9], and ensures the platform avoids local minima. This global path estimate provides waypoints as sub-goals to the local planner. The local planner then computes the best local path and provides a control signal to the robot’s low-level controller for navigation.

The local planner—denoted Multiple Steer Drive Unit Local Planner, or MSDU—is evaluated on a real-world robotic platform. It is demonstrated to provide more accurate positioning than existing local planners, achieving a smaller distance from the desired goal position in terms of both translation and angular displacement, while also taking a shorter path (in terms of both translational and angular distance) to the goal.

## 2. Related Work

This paper focuses on the problem of motion planning and trajectory optimization. The challenge of motion planning is to produce paths or trajectories for a robot to travel to a specified goal, where a path is a curvature defined as a continuous or discretized function and a trajectory also contains velocity information. Trajectory optimization is often difficult as obstacles and complex robot dynamics have to be considered. The motion planner needs to consider both kinematic and kinodynamic feasibility. A kinematically feasible output simply means that the output paths are possible to drive: for example, the paths have continuous curvature. A kinodynamically feasible output is one where the given velocity and acceleration bounds on the vehicle are met.

Local motion planners that handle short-term planning with a limited horizon are used in complex environments where obstacle avoidance is the key. One “classical” local planner is the Dynamic Window Approach (DWA) [14], which, given a goal and current sensory readings, searches for feasible velocities that will drive the robot towards the goal. Another approach that is commonly used in practice is based on the Elastic-Band (EB) planner [15]. However, these planners provides outputs that are not kinodynamically smooth and lack a velocity profile [10]. Other approaches based on potential fields have also been proposed [16].

The elastic band has been extended with an optimization framework TEB [9,17] and provides a local planner that utilize optimization, which provides a trajectory that is followed by a separate controller. TEB is designed for differential drive, car-like and full holonomic vehicles. In [9], a set of possible local trajectories were used blending the difference between global and local planning. TEB-based formulations have also been utilized for motion planning of manipulators [18].

There is a large variety of optimization-based approaches which, nowadays, are becoming more and more popular primarily thanks to the increased computational power available. Often the task of trajectory generation and tracking/control is separated into two different entities [19,20]. As trajectory optimization is essentially what is solved in the MPC scheme and for example, similar to the approach suggested here, Schoels et al. [10] is a mixture of these two as the proposed system both generates trajectories and performs the tracking. In general, a more complicated footprint in combination with limited dynamics of a platform makes both the global motion planning search part as well as the trajectory generation and tracking more difficult, hence the need to separate them.

As the output of a global planner is typically a path, we can also define approaches that generate trajectories to reach points along the route as “path following” approaches. A generic description of wheeled platforms and path following is described by Oftadeh et al. [11] which also contains evaluations of a combined steer and drive platform. One key difference here is how the controllers are coupled with the error directly between the current pose and corresponding static path and do not rely on horizon-based optimization, and the followed path is assumed to be free of obstacles. There are a large variety of path-following for cars, such as lateral control, which also is formulated for 4WS4WD (four-wheel-steering–four-wheel-driving) or 4WID (four-wheel-independent-drive) [21,22] and often focuses on more complex dynamics and stability at higher speeds and not on obstacle handling. Typically these car-like vehicles have a limitation on the amount of steering each wheel can undertake [23].

How to formulate obstacle handling efficiently into an optimization framework is of importance and is related to both how to represent the vehicle’s footprint, but also how to represent the shape of the obstacles. Circular or ellipsoidal constraints is commonly used [24] but other techniques using, for example, convex polygons described through signed distance functions [10,24].

Depending on the kinematic configuration and footprint of the platform there are many different motion planners, such as RRT [25], probabilistic roadmaps [26] and lattice-based motion planners [19,27]. There are also many variants of the RRT planner, such as RRT* [28] which is provably asymptotically optimal, RRT-Blossom [29] which is designed for highly constrained environments or RRT*-UNF [30] which targets dynamic environments.

As the given platform has a rather simplistic footprint and good maneuverability, we exploit this to have a fast Dijkstra-based global planner that is fast to compute and therefore can be continuously re-invoked with an up-to-date representation containing dynamic and static obstacles.

## 3. System Design

This section introduces the system that the MSDU Local Planner is designed to integrate with, including the robotic platform, the available sensors, the existing planning and navigation capability and the low-level control functionality. The challenges associated with the pseudo-omnidirectional motion capabilities of the platform are presented in Section 4, which describes the geometry and kinematics of the combined steer and drive wheels, and how the velocity and control values can be suitably represented to provide constraints to the optimization formulation. Section 5 then introduces the MSDU Local Planner presented in this paper with Section 5.1 describing the formulation of the optimization problem.

### 3.1. The Platform

The robotic platform used in this work is presented in Figure 1. It has four steerable wheels located in each corner and is what is often termed as a “pseudo-omnidirectional” platform [16].

The steerable wheels in the steerable wheel research platform are an in-house design and are based on the outrunner hub-motors design commonly used in commodified personal e-mobility vehicles. These motors are superior when it comes to the torque-to-cost ratio, which was utilized to realize a direct-drive (no gearbox) solution for both steering and driving motion. High-quality motion performance is obtained using high-accuracy position sensors and state-of-the-art motor control. The result is a compact, high-performance, reliable and very quiet actuation module.

The vehicle control is designed to receive independent linear velocities in the 2D plane and rotation ($v_{x},v_{y},\omega$). To ensure high-performance dynamics of the vehicle-motion reference tracking, a dynamic model is used to compute feedforward torque to joints in a prediction-closed loop correction scheme. A dedicated Kalman filter is employed to make the vehicle control tolerant to potentially relatively low frequency and jittery references coming from the motion planner in the transition to the high frequency real-time system. In the tests reported in the present paper, the vehicle control system is reading the communication bus for new references from the planner at 500 Hz.

The wheel odometry computation is designed to take advantage of the redundancy of this specific configuration of four steerable wheels such that it selects wheels to be included to compute the measured 2D pose based on the motion. For example, any wheel that is close to ICR is omitted from the odometry calculation. This results in an accurate and robust measured vehicle-speed estimation, which is communicated to the navigation system at 500 Hz.

### 3.2. Steerable Wheels

The wheel configuration of multiple combined steer and drive wheels units was selected as it can provide omni-motion capability, while also having other benefits relative to other comparable systems:*Payload capability.* A steerable wheel is straightforward to scale in payload.*Stability and stiffness.* A steerable wheel allows stiff or no suspension, enabling a minimum footprint for a given mobile manipulation application stiffness requirement.*Wheel odometry.* Steerable wheels enable accurate wheel odometry due to their limited slippage. For robots with more than two steerable wheels the system also has inherent redundancy that enables diagnostics of vehicle motion estimation based on wheel motions.*Robustness.* Steerable wheels are relatively robust to floor and environment conditions, and vehicle motion control is maintained even if some wheels lose floor contact, as long as at least two wheels remain in contact with grip. The size and material of the wheel can be chosen to suit the application; for example, a larger- or smaller-sized wheel can be used, and the system can employ soft material for the tire to increase the contact area if pressure in the floor contact needs to be limited.*Motion performance.* Steerable wheels are capable of high acceleration, deceleration, and turning due to good ground grip and low steering inertia. Steerable wheels have high scalability of speed capacity, and low vehicle-direction isotropic friction losses or slippage.*Flexible wheel configuration.* Steerable wheels can be placed arbitrarily in the chassis for design flexibility, and the number of wheels can be arbitrarily chosen. This flexibility in the wheel configuration can be useful to allow for modular robot payload and motion performance capacity.

Note that having more than two combined steer and drive wheels will provide redundancy to the system that can be beneficial for better traction in general and better handling of non-flat surfaces.

### 3.3. Perception

The platform is equipped with a SICK TiM571 2D LiDAR, which is used for localization, navigation, and planning, as well as for obstacle detection. It is also used for building the map (see Figure 8) used by the robot to perform localization, navigation, and planning. The sensor characteristics taken from the data sheet are a systematic errors of ±60 mm and statistical errors of ±20 mm at ranges up to max range of 25 m with 90% remission and for 10% remission up to 6 m range the statistical error is ±10 mm; however, all numbers are typical and dependent on ambient conditions.

### 3.4. Localization, Navigation, and Planning System

The platform is equipped with all systems required for navigation; localization, global planning and the proposed MSDU local planning (where MSDU stands for Multiple Steer-And-Drive Unit) that provides the control output. All system components exist as ROS [31] packages and the proposed local planner is compatible with the navigation stack [32]. When provided with a map of its environment, the robot can localize within its map, using its range sensor and odometry readings. As a localization module we utilize the AMCL package with only minor parametric changes; the motion model is set to “omni-corrected”. The robot also has a global planner that can provide a path based on the robot’s map of the environment and also incorporated LiDAR readings. The global planner we use is the planner provided in the navigation stack with default parameters. The output is a path containing a set of poses and does not contain any velocity information.

The platform footprint is encoded as two circles. One circle has an inscribed radius, which gives the smallest opening you could drive the robot through with a *specific* angle, and the second circle has a circumscribed radius which is the smallest opening you could drive the robot through given *any* angle. What this shows us is that the global planner does not fully factor in the shape of the platform and it is up to the local planner to handle narrow passages where the opening size is between the inscribed and circumscribed radii, as can be seen in Figure 16.

Furthermore, the global planner does not consider the full kinematics of the robot and the generated global path is non-smooth. However, the global planner does consider the surrounding using LiDAR data (through the local costmap) and the given map (global costmap) to rapidly compute paths that keeps a distance from obstacles and that find plans that are suitable from a global perspective.

The local planner presented in this paper is designed to integrate with the global planner and use the provided path to generate waypoints as input to the local planner, which then formulates a control output for the low-level wheel controller. Unlike the global planner, the local planner factors in a large number of constraints, including the kinematic constraints of the robot and a footprint that can handle different orientation (to ensure safe navigation through confined spaces). Furthermore, the local planner integrates real-time information from the LiDAR sensor to guarantee that the generated control actions provide motions that are collision free.

Figure 2 presents an overview of the different components of the system and how they interact. As can be seen, the local planner integrates information from many components, including the global planner and the sensor data from the LiDAR, to calculate and output a control signal to the robot.

## 4. Geometry and Kinematics of the Combined Steer and Drive Wheels

This section provides an overview of the geometry and kinematics of the combined steer and drive wheels present on the robotic platform used in this paper. It outlines the physical limitations on the linear and angular velocities of the wheels, which will be used to provide the constraints for the optimization formulation in Section 5 below. Please note that we do not consider nor model any wheel slippage.

### 4.1. Definitions

Let the input control signal to the platform be denoted $u_{p}=\left(v,\omega \right)$ where $v=(v_{x},v_{y})$ is the linear velocity (a 2D vector) and ω is the rotational velocity (a scalar). Using a 2D world reference frame {A}, let θ denote the orientation of the platform and x,y its 2D position (see Figure 3a). Then we have the following equations for the velocity of the platform in the world frame:(1)$$\begin{array}{cc} {\dot{x}}^{\left\{A\right\}} & =v_{x}^{\left\{A\right\}}\\ {\dot{y}}^{\left\{A\right\}} & =v_{y}^{\left\{A\right\}}\\ {\dot{\theta }}^{\left\{A\right\}} & =\omega \end{array}$$

We also have the platform frame {B}, which has its origin at the center of the platform. The {B} frame is defined so that the *x* direction will point “forward” (see Figure 3b). If the velocity is defined in the platform frame {B}, the resulting equations become:(2)$$\begin{array}{cc} {\dot{x}}^{\left\{A\right\}} & =v_{x}^{\left\{B\right\}}\cos\left(\theta \right)-v_{y}^{\left\{B\right\}}\sin\left(\theta \right)\\ {\dot{y}}^{\left\{A\right\}} & =v_{x}^{\left\{B\right\}}\sin\left(\theta \right)+v_{y}^{\left\{B\right\}}\cos\left(\theta \right)\\ {\dot{\theta }}^{\left\{A\right\}} & =\omega \end{array}$$

### 4.2. Wheel Configuration

On the platform there are four combined steer and drive wheels ( $W_{1\ldots 4}$). We denote for each wheel $W_{i}$ the velocity $v_{i}$ and the steering angle $\varphi_{i}$. The position of the wheel $p_{i}^{\left\{B\right\}}=\left(p_{x}^{i},p_{y}^{i}\right)$ is given in the platform frame. The velocity vector $v_{i}$ is used to represent the combination of $\varphi_{i}$ and $v_{i}$ (see Figure 4).

### 4.3. The Effect of the Control Input on the Wheels

Given a control value u=(v,ω) to the platform, it is necessary is to compute the corresponding steering angle $\varphi_{i}$ and drive wheel velocity $v_{i}$ of each wheel.

We therefore seek a function *f* with the following property:(3)$$\begin{array}{c} \left(\varphi_{i},v_{i}\right)=f\left(p_{i}^{\left\{B\right\}},v^{\left\{B\right\}},\omega \right). \end{array}$$

In this formulation, v (and p) are expressed in the platform frame {B}.

Intuitively, function *f* consists of two different components. (1) $f_{v}$ which handles linear motion and (2) $f_{\omega }$ which takes care of rotational motion. The function is a combination of the two: “$f=f_{v}+f_{\omega }$”.

#### 4.3.1. Linear Velocity Component $f_{v}$

Firstly, assume there is only a linear control component v (and ω=0). This would correspond to each wheel having the same steering angle $\left(\varphi_{1}=\varphi_{2}=\varphi_{3}=\varphi_{4}\right)$, given by the velocity vector v as:(4)$$\begin{array}{cc} \left(\varphi_{i},v_{i}\right) & =\;f(p_{i},v,0)\\ \varphi_{i} & =\;atan2(v_{y},v_{x})\\ v_{i} & =\;∥v∥ \end{array}$$

Note that: (1) the steering angle is independent on the wheels’ position $p_{1\ldots 4}$, and (2) when there is no velocity v given in this case, the steering angles $\varphi_{1\ldots 4}$ can be arbitrarily set.

#### 4.3.2. Rotational Velocity Component—$f_{\omega }$

Now, assume that there is only a rotational velocity control component ω (and v=0). Given that ω≠0 the steering angles can be computed as (see also Figure 5):(5)$$\begin{array}{cc} \left(\varphi_{i},v_{i}\right) & =\;f(p_{i},0,\omega )\\ \varphi_{i} & =\;atan2\left(p_{y}^{i},p_{x}^{i}\right)+\frac{\pi }{2}\\ v_{i} & =\;\omega \times p_{i}\\ v_{i} & =\;∥v_{i}∥ \end{array}$$
where for the × computation we have assumed that $\omega ={\left[0\;0\;\omega \right]}^{T}$ and $p_{i}={\left[p_{x}^{i}\;p_{y}^{i}\;0\right]}^{T}$. Note that $v_{i}$ is a vector (compared to $v_{i}$ which only is a scalar).

Note that if there is a wheel located at the center of the platform frame ($p_{x}^{i}=p_{y}^{i}=0$), this wheel angle is not defined and could in principle be set arbitrarily; note that this point is the Instantaneous Center of Rotation (ICR).

#### 4.3.3. Combining Linear and Rotational Components

From the previous sections we have obtained a set of linear velocity vectors (v and $v_{1\ldots 4}$), where v originates from the linear velocity component $f_{v}$ and $v_{1\ldots 4}$ from $f_{\omega }$.

These velocity vectors are combined through vector summation as follows:(6)$$\begin{array}{cc} \left(\varphi_{i},v_{i}\right) & =\;f(p_{i},v,\omega )\\ \varphi_{i} & =\;\underset{——}{/\omega \times p_{i}+v}\\ v_{i} & =\;∥\omega \;\times \;p_{i}+v∥ \end{array}$$

Note that if $v=-\omega \times p_{i}$, we have the point of rotation (ICR) at $p_{i}$. Note at this exact point the wheel can in principle be set arbitrarily, however, this point is singular as small changes to the ICR will require rapid changes to the angle φ. In short, this singular point can be avoided by ensuring that the ICR point is sufficiently distant from $p_{i}$.

The ICR point is a function of the control $u_{p}=\left(v,\omega \right)=\left(v_{x},v_{y},\omega \right)$ as:(7)$$\begin{array}{cc} {\mathrm{ICR}}_{x} & =-\frac{v_{y}}{\omega }\\ {\mathrm{ICR}}_{y} & =\frac{v_{x}}{\omega } \end{array}$$

Note that if there is no rotational element ω, the ICR will be at infinity.

### 4.4. Representing the Control Action

Taken from the previous sections we have the following ODE:(8)$$\begin{array}{cc} {\dot{x}}^{\left\{A\right\}} & =v_{x}^{\left\{B\right\}}\cos\left(\theta \right)-v_{y}^{\left\{B\right\}}\sin\left(\theta \right)\\ {\dot{y}}^{\left\{A\right\}} & =v_{y}^{\left\{B\right\}}\sin\left(\theta \right)+v_{y}^{\left\{B\right\}}\cos\left(\theta \right)\\ {\dot{\theta }}^{\left\{A\right\}} & =\omega \end{array}$$

We need to incorporate the different wheel configurations; that is, to add the different wheels into the formulation. From the discussion above, there are limitations on the turning of each wheels as well as singular points. The key question here is how can we constrain the control to ensure that it is feasible. Equation (Equation 6) above is a function that satisfies the property stated in Equation (Equation 3):(9)$$\begin{array}{cc} \left(\varphi_{i},v_{i}\right) & =f(p_{i},v,\omega ) \end{array}$$

That is, Equation (Equation 6) computes the orientation and velocity of each wheel. However, it is necessary to ensure that the velocity $v_{i}$ of each wheel is within bounds and does not exceed a max value. Similarly there is also a limitation on the turning rate of each wheel $\dot{\varphi_{i}}$.

The first step is to find a more suitable representation of the control $u_{p}$ that is better aligned with the wheel configuration such as the steering angle of the wheels. Therefore we utilize the following control variable as proposed in [33]:(10)$$\begin{array}{c} u_{p}=\left(v,\varphi ,\omega \right) \end{array}$$
where $v=\sqrt{v_{x}^{2}+v_{y}^{2}}$ and $\varphi =atan2(v_{y},v_{x})$. In essence we change the base of the linear velocity to contain the linear velocity direction φ, as well as the linear velocity along this direction as *v*. As it is easy to convert between the different linear velocity representations ($v_{x}=v\sin\left(\varphi \right)$, $v_{y}=v\cos\left(\varphi \right)$) they are used interchangeably in this work.

If we now only have a linear velocity (ω=0) each steering wheel angle is given by φ, that is $\varphi_{i}=\varphi$ for each wheels *i*. Given this representation we can now add limitations on the change of the control $\dot{\varphi }$ to better reflect the limitation on the change in steering angle. One key situation is when approaching the goal: the control action must be limited, otherwise small changes in linear velocities would require extremely large changes in wheel steering angles $\dot{\varphi_{i}}$. Note that this is not the same problems as discussed above regarding ICR as when ω=0 the ICR is at the center of the platform and far from the wheels’ locations $p_{i}$.

Another benefit is that the change in linear velocity control representation is that the linear velocity speed is separated from the orientation components. This allows a more intuitive way of formulating an acceleration profile. Furthermore, a profile on the change of linear direction—which is highly influential of the steering wheel angles—can be formulated. Specifically, the following limitations are added on the control variables:(11)$$\begin{array}{cc} -{\dot{v}}^{\max} & \leq \dot{v}\leq {\dot{v}}^{\max}\\ -{\dot{\varphi }}^{\max} & \leq \dot{\varphi }\leq {\dot{\varphi }}^{\max}\\ -{\dot{\omega }}^{\max} & \leq \dot{\omega }\leq {\dot{\omega }}^{\max} \end{array}$$

Note that these are not directly connected to any physical limitation of the steer and drive wheels. For example, the change in steering wheel angle is often fast (in the platform used in the evaluation the changes are >20 rad/s). However, these limitations are useful for a smooth drive characteristic.

There is also a maximum velocity limitation on the drive wheel. This limitation is also explicitly considered and is detailed in the next section.

### 4.5. Limitations on the Control Action Due to Maximum Velocity of the Drive Wheels

The linear and angular velocity of the wheels are connected. Informally speaking, you cannot both drive quickly and turn quickly at the same time, because, as with differential drive platforms, the same wheel velocities $v_{i}$ are utilized to both obtain the linear velocity v as well as the angular velocity ω of the platform. However, for a differential drive platform the ICR point is on the line that connects the left and right wheel, while for the platform used here the ICR can be set arbitrarily. The maximum rotational velocity a wheel can achieve without having any linear velocity (that is the ICR is given at (0, 0)) is given by:(12)$$\begin{array}{c} \omega_{i}^{\max}=\frac{v_{i}^{\max}}{\left|\right|p_{i}\left|\right|} \end{array}$$

It is clear that the rotational speed is greatly dependent on the distance between the ICR and the wheels. For a differential driven platform with a linear velocity forward and turning right it will be the left wheel that will reach the $v_{i}^{\max}$ boundary first. Hence it is the wheel with the furthest distance from the ICR that will be the limitation. In principle for the platform at hand, it would be possible to have a larger linear velocity $\left(v_{x},v_{y}\right)=\left(v,0\right)$ (along the *x*-axis) than $\left(v_{x},v_{y}\right)=\left(\sqrt{2},\sqrt{2}\right)$ as the distance between the ICR and the wheel furthest away is longer in the latter case.

In the proposed approach this difference is neglected and we consider a combined linear and rotational max boundary as follows:(13)$$\begin{array}{cc} -\omega^{\max} & \leq \omega -\frac{v}{d}\leq \omega^{\max}\\ -\omega^{\max} & \leq \omega +\frac{v}{d}\leq \omega^{\max} \end{array}$$
where $d=d_{i}=\left|\right|p_{i}\left|\right|$ which is assumed to be the same distance for all wheels *i*. In practice, this corresponds to always assuming the worst-case scenario where one wheel is always the furthest possible distance away from the ICR.

## 5. Defining the Optimization Problem

This section outlines the non-linear optimization problem that forms the basis of the MSDU Local Planner. The core objective of the MSDU Local Planner is to obtain a local feasible trajectory that drives the platform towards a goal. This goal is obtained from a global planner that is periodically updated (in the order of approx. 1 Hz). Hence, the local plan obtained does not have to consider getting stuck despite its local nature.

We here formulate a non-linear optimization problem that will generate a feasible trajectory. Due to the computational complexity that comes with non-linear solving, care has to be taken to formulate an optimization problem that is fast enough to solve. This will impact how the problem is formulated, as well as restricting the size of certain parameters; for example, the look-ahead distance and the sampling resolution. On the positive side, the non-linearity approach allows us to be much more free in how we select the objective function.

### 5.1. Problem Formulation

Two different approaches are used to steer the trajectory generation by the optimization. The first approach is to add factors into an objective function. The second is to post constraints on the different variables. Constraints are a very powerful and intuitive way to steer the optimization, but can be problematic if constraints are posted that simply cannot be satisfied. One example would be the goal pose $P^{goal}$ that we would like to arrive at. Typically, it is not guaranteed that given other constraints on, for example, the acceleration, velocity, and turning speed limits, that we can reach the goal. Instead, we use the distance between the current pose and future poses to the goal pose in the objective function. If the goal cannot be reached, that is acceptable as no constraints are violated and at the same time the minimization of the cost function (that is, the distance to the goal) will drive the vehicle towards the goal.

The state s consists of the vehicle pose (x,y,θ) and linear velocity, direction and angular velocity (v,φ,ω) whereas the optimization control variables u are (dv,dφ,dω). Note that the control action $u_{p}$ used to drive the vehicle is actually part of the state, however, we will use an additional optimization variables of derivatives of the control values to fulfill additional requirements such as limiting the maximum acceleration permitted; see Equation (Equation 10).

The model of the vehicle dynamics ($\dot{s}=f\left(s,u\right)$) is described as:(14)$$\begin{array}{cc} \dot{x} & =v\cos(\varphi )\cos(\theta )-v\sin(\varphi )\sin(\theta )\\ \dot{y} & =v\cos(\varphi )\sin(\theta )+v\sin(\varphi )\cos(\theta )\\ \dot{\theta } & =\omega \\ \dot{v} & =dv\\ \dot{\varphi } & =d\varphi \\ \dot{\omega } & =d\omega \end{array}$$

Given the dynamics, we formulate a constrained optimal control problem (OCP):(15)$$\begin{array}{cccc} \underset{s,u}{\mathrm{minimize}} & \phi \left(T\right)+\int_{0}^{T}l\left(s\left(t\right),u\left(t\right)\right)dt\\ \mathrm{subject}\mathrm{to} & s(0) & ={\hat{s}}_{0}, & \\ & \dot{s}\left(t\right) & =f(s(t),u(t)), & t\in [0,T],\\ & h(s(t),u(t)) & \leq 0, & t\in [0,T],\\ & d(s(t),o,c) & \leq 0, & t\in [0,T],o\in O,c\in C \end{array}$$
where *T* is the horizon length in seconds, ϕ(T) is the terminal cost, *l* is the cost for time *t*, ${\hat{s}}_{0}$ is the initial state, *f* is the vehicle dynamics function (Equation (Equation 14)), *h* is the path constraints containing limits on inputs u, such as max accelerations, as well as pure state constraints on s, such as bounds on max velocities, and finally *d* provides a mean to ensure collision free state poses given a set of obstacle points O as well as a set of circles C representing the shape of the vehicle. Both the obstacle point $o=[o_{x},o_{y}]$ and the circle $c=[c_{x},c_{y},c_{R}]$ are given in the vehicle frame where $c_{R}$ is the radius of the circle.

To solve the OCP problem defined above we discretize it into a non-linear program (NLP) using multiple shooting [34]. The trajectory consists of vehicle states at discrete timestamps and holds *N* steps covering *T* seconds which gives us that each increment brings us $dt=\frac{T}{N}$ seconds into the future.

The discrete decision variable is $\zeta ={\left\{s_{i},u_{i}\right\}}_{i=1}^{N}$, and the non-linear program is written as:(16)$$\begin{array}{cccc} \underset{\zeta }{\mathrm{minimize}} & \phi \left(\zeta_{N}\right)+\sum_{k=0}^{N-1}l\left(\zeta_{k}\right)\\ \mathrm{subject}\mathrm{to} & s_{0} & ={\hat{s}}_{0}, & \\ & s_{k+1} & =F(s_{k},u_{k},dt), & k=0\cdots N-1\\ & h(s_{k},u_{k}) & \leq 0,\; & k=0\cdots N\\ & d(s_{k},o,c) & \leq 0, & k=0\cdots N,o\in O,c\in C \end{array}$$
where the objective function is described in Equation (Equation 19), *F* is the discrete model of the dynamics (see Equation (Equation 20)), the path constraints *h* are given in Equations (Equation 21) and (Equation 22) and finally the constraints to ensure collision-free motions *d* are described in Equation (Equation 24).

As we are primarily interested in the next control action to take, we follow the classical model-predictive control (MPC) scheme and use the obtained decision variables ζ to extract the next control action. Depending on inherent lag in the system, it is also possible to take not just the first control action available but to take a future one.

Because the problem is formulated as a standard non-linear program, it can be straightforwardly integrated into existing non-linear solvers. In this work, the formulation was implemented with CasADi [35] which here utilizes the Ipopt library [36] to solve the posted non-linear problem.

The objective and constraints are discussed further in the following sections.

### 5.2. Inputs and Outputs

As described above we continuously receive a global plan (at approximately 1 Hz) from which we extract the next local “goal” based on our current localization estimate, which is also provided continuously (at 50 Hz, where the localization system runs slower and is dependent on the translation and rotational distance, but is augmented with odometry readings which are obtained at 50 Hz). To simplify the formulation the local goal is converted into the robot frame {B}, which allows us to assume that we always start at pose (0, 0, 0). The continuous sensory input (at 10 Hz) is already provided in the robot frame {B} as the sensory setup is located on the robot itself.

The controller or the local planner is queried at 10 Hz in which the latest received plan, localization estimate, and sensory data are used.

The output is the next control action to be executed $u_{p}=\left(v_{x},v_{y},\omega \right)$.

### 5.3. Objectives

As discussed above, the force that drives the robot towards the goal $g=(g_{x},g_{y},g_{\theta })$ lies in the cost objective which contains the distance between the goal and each pose in the *N*-step long trajectory ${\left(x,y,\theta \right)}_{1\cdots N}$. The goal part to the objective is as:(17)$$\begin{array}{cc} J^{goal} & =\sum_{i=1}^{N}w_{i}^{x}{\left(g_{x}-x_{i}\right)}^{2}+w_{i}^{y}{\left(g_{y}-y_{i}\right)}^{2}+w_{i}^{\theta }{\left(g_{\theta }-\theta_{i}\right)}^{2} \end{array}$$
where we have different weighting factors $w_{x}^{1\cdots N}$, $w_{y}^{1\cdots N}$ and $w_{\theta }^{1\cdots N}$. These weights can be selected and tuned as needed, but in the evaluation presented in this paper all position weights were set to be the same. It is also possible to have a lower cost on the intermediate weights in the range (1…N−1) compared to the last terminal state weights $w_{N}$. The core idea is that we want to steer the optimization towards the goal as quickly as possible. Additional constraints to limit the velocities and accelerations were also added as discussed in Section 5.4 below.

Another cost relates to the magnitude of the control actions utilized. This was found particularly important to limit the amount of turning when driving the platform close to the goal. The cost on the decisions variables related to the derivatives of the generated control output is defined as:(18)$$\begin{array}{cc} J^{control} & =\sum_{i=1}^{N}w_{i}^{dv}{\left(dv\right)}^{2}+w_{i}^{d\varphi }{\left(d\varphi \right)}^{2}+w_{i}^{d\omega }{\left(d\omega \right)}^{2} \end{array}$$

Our objective is the sum of the above and can be rewritten as:(19)$$\begin{array}{cc} \phi \left(\zeta_{N}\right)+\sum_{k=0}^{N-1}l\left(\zeta_{k}\right)= & \sum_{i=1}^{N}x_{i}^{T}Q_{i}x_{i}+\sum_{i=1}^{N}u_{i}^{T}R_{i}u_{i} \end{array}$$
where $x_{i}={\left[g_{x}-x_{i},g_{y}-y_{i},g_{\theta }-\theta_{i}\right]}^{T}$ and $Q_{i}$ together with $R_{i}$ are diagonal weighting matrices.

### 5.4. Constraints

The constraints added relate to limitations of the vehicle, such as maximum drive velocities, but also define harder acceleration limits to obtain a softer and smoother driving characteristic. The cost objective defined in Section 5.3 pulls all states towards the goal, while the constraints limit the velocity profile. The constraints are also used to handle obstacle avoidance by the planner.

#### 5.4.1. Velocity Profiles and Vehicle Constraints

The constraints utilized are to ensure strict boundaries as max velocities and accelerations are not exceeded. Constraints are also used to ensure that the generated trajectory is consistent; that is, that the states’ progression by applying the corresponding control action will bring the current state to the next. In principle it would be sufficient to provide the initial state and the generated control action to derive the trajectory, but this work utilizes a multiple-shooting [34] approach where we keep track of the states in each iteration which is useful as some constraints added contains state information (note that the control output sent to the vehicle is defined in the state).

Given a control action u=[dv,dφ,dω] the next state is computed by integrating the vehicle dynamics *f* specified in Equation (Equation 14), using Runge–Kutta RK4 for the integration. The equality constraint used to make sure the state integrated with control is consistent with the next state $s_{k+1}=F\left(s_{k},u_{k},dt\right)$ is defined by: (20)$$\begin{array}{cc} k_{1} & =f(s_{k},u_{k})\\ k_{2} & =f(s_{k}+k_{1}\frac{dt}{2},u_{k})\\ k_{3} & =f(s_{k}+k_{2}\frac{dt}{2},u_{k})\\ k_{4} & =f(s_{k}+k_{3}dt,u_{k})\\ s_{k+1} & =s_{k}+\left(k_{1}+2k_{2}+2k_{3}+k_{4}\right)\cdot dt \end{array}$$

To limit the control actions as well as the maximum velocities, the following non-equality constraints are added; also see Equation (Equation 11):(21)$$\begin{array}{cc} -dv^{\max}dt & \leq dv\leq dv^{\max}dt\\ -d\varphi^{\max}dt & \leq d\varphi \leq d\varphi^{\max}dt\\ -d\omega^{\max}dt & \leq d\omega \leq d\omega^{\max}dt \end{array}$$

To ensure that the maximum drive velocity on each wheel is not exceeded, the following constraints are utilized:(22)$$\begin{array}{c} -\omega^{\max}\leq \omega -v/d\leq \omega^{\max}\\ -\omega^{\max}\leq \omega +v/d\leq \omega^{\max} \end{array}$$

Here, *d* is the distance between the centre of the platform and the wheels, which is assumed to be the same for all wheels.

The above two sets of constraints are used on all control variables in the optimization.

#### 5.4.2. Obstacle Constraints

As one of the objectives is to drive in confined areas it is necessary to have good spatial resolution of obstacles. Rather than using a grid-based representation that would discretize the environment and hence lower the available resolution, instead we use range readings directly to have as high resolution as possible and to receive quick feedback. This requires that we have a good overview of the surrounding of the platform at all times. In essence, an obstacle $o=[o_{x},o_{y}]$ is a point in 2D.

Note that the global path planner will, to a great extent, handle obstacle avoidance while providing the global path. However, the obstacle constraints are necessary to safely drive at close proximity to obstacles.

From the raw range readings we process the data to only extract the key LiDAR readings. This is essential as each constraint will add additional complexity to the optimization problem at hand. How the obstacle constraints are formulated is also greatly affected by the representation used to model the platform. Due to the square shape of the platform the orientation is essential. For example, to drive through a narrow passage straight ahead along the x-axis, the orientation θ must be kept close to the following angles (0,π/2,π,3π/4) as the required width would be approximately $\sqrt{2}$ larger if we instead had an orientation of π/4.

As the optimization problem can be altered on the fly, there are two different shapes that are used. One is used for driving, and thus we want the platform to drive with a specific forward direction, and the other is used for close-proximity driving when reaching goals. In the first scenario we define the footprint as two circles, one slightly forward and one slightly backwards. The front circle and back circle will stick out and make the footprint form in a clear direction, which is the best way to pass a narrow passage. The front circle will act as a “plow” that divides obstacles on either side. For close proximity near a goal, we instead approximate the shape of the robot better using five circles, one center and the remainder used to cover the corners of the platform.

To approximate the platform using only a circle would simplify the computation of the constraint to be the following:(23)$$\begin{array}{c} \sqrt{{\left(x_{k}-o_{x}\right)}^{2}+{\left(y_{k}-o_{y}\right)}^{2}}\geq R\\ \;\forall \;o\in O \end{array}$$
where *R* is the radius of the circle. This constraint is added both for the different states k=1…N as well as for all obstacles o∈O, which gives us in total of N×|O| where |O| is number of obstacles.

Circles that have have their centres at an offset ($c=[c_{x},c_{y},c_{R}]$) also require that the platform orientation θ is considered. These obstacle constraints are formulated as:(24)$$\begin{array}{c} \sqrt{{\left(x_{k}+c_{x}\cos\left(\theta_{k}\right)-c_{y}\sin\left(\theta_{k}\right)-o_{x}\right)}^{2}+{\left(y_{k}+c_{x}\sin\left(\theta_{k}\right)+c_{y}\cos\left(\theta_{k}\right)-o_{y}\right)}^{2}}\geq c_{R}\\ \forall \;o\in O,\forall \;c\in C \end{array}$$
which gives us the total number of constraints added to be N×|O|×|C|, where |O| and |C| is the number of obstacles and circles respectively.

One difficulty not addressed up to this point is that each obstacle o is directly derived from sensory data and hence is subject to various sets of noise. If we are operating the robot close to an object it might happen that in one iteration the obstacles are on the “correct” side of the boundary, whereas in the next iteration they are not and the optimization problem is infeasible. To handle this situation we avoid constraints using the first state variable (k=0) as this state is an initial condition and it is not possible to alter it in the optimization. In principle, one could also remove collision checks on further states ahead that then could handle larger noise values or cause the robot to back off when other dynamic entities such as people within the collision boundaries. As the trajectory is discretized based on dt, ignoring any step apart from k=0 is not recommended especially if dt is large as not all actions effect the state and its collision boundaries will be incorporated and checked. If there is no feasible solution the vehicle is stopped; in practice this can happen if you quickly place an obstacle into the collision constraint zone, then the vehicle will simply stay still until this obstacle is removed.

Another approach to handling the noisy LiDAR data would be to incorporate the obstacle avoidance as part of the objective function. One option here would then be to utilize sigmoid functions that give a high cost if the the obstacle is within the collision boundary and a low one if it is not. One problem with this is that from an optimization point of view you also want an objective that is smooth (i.e., has a nice derivative) which from a numerical and practical point of view could be challenging. Due to this, we instead opted for using constraints on the obstacle avoidance and we found this approach to be more intuitive.

#### 5.4.3. Constraints to Avoid Singularities

As described in Section 4.3.3 there is a singular configuration when the ICR is very close to the wheel position which makes small changes in linear and angular velocities (v,ω) to create large changes in the steering angle of that wheel. To avoid control actions that are given in these singular regions, one option is to adopt the following set of constraints (one for each wheel position $p_{1\cdots 4}$):(25)$$\begin{array}{c} ||\mathrm{ICR}-p_{i}\left|\right|\leq R \end{array}$$
where *R* is the boundary radius that the ICR should not enter. However, while this constraints does seem to address the problem, the time complexity involved in adding these constraints was simply not worth it compared to using a more simplistic approach. Instead of adding this limitation into the optimization problem we can simply adjust the corresponding command sent to the platform (or have the platform do this internally) by adjusting the command u=(v,ω) to make sure that the above criteria $||\mathrm{ICR}-p_{1\cdots 4}\left|\right|$ holds by adjusting the control values; see Figure 6.

This can be achieved quickly by first checking if the ICR is within any of the bounds. If not the command is valid and can be directly sent to the low level controller. If it is within any of the bounds we have the option of changing either the linear or angular velocity component or both before sending the command. Here we adopted the approach to adjust the linear v component and to keep the rotational velocity ω. One benefit of doing this at a higher level compared to just sending any control commands and letting the low-level controller handle it is that the the control values that will be used in the forward model will be more correct. It should also be noted that the optimization problem formulated here is to be used as a tracking-based controller as the key input is the next local goal pose, rather than a trajectory to follow, which makes the system robust to large disturbances.

Performing the navigation task though the doorway, as depicted in Figure 16 using ICR as constraints into the optimization had an average optimization time of 46±29 ms compared to 29±3 ms if ICR were instead corrected after optimization as described in the previous paragraph. It is worth noting the large variance which indicates that optimization problem is rather ill-posed but yet possible to solve.

#### 5.4.4. Initial Conditions

As mentioned above, the initial state pose is assumed to be (0,0,0) and the local goal g is given in the robot frame {B} along with the extracted obstacles $o_{1\cdots O}$ from LiDAR data. One part that has not been addressed is how the remaining state variables, that from a control perspective is actually used to command the platform at hand, are determined. Looking at the state Equation (Equation 14), we also have the linear velocity *v*, the linear velocity direction φ and the angular velocity ω. One key question is how to obtain and provide these values.

As the platform is non-holonomic by nature, since there it takes some time in order to turn the steering wheels, we cannot directly set any arbitrary φ value. Nor can we directly set an arbitrary rotational velocity ω and expect an instantaneous response. One approach could be to define that the robot always starts moving in one direction and then use this as an initial condition, but this would limit the advantages of the platform that you can start moving in an arbitrary direction or to start by a rotation. Instead of modeling the time taken for the robot to rotate its wheels which corresponds to the commanded linear and angular velocity, we at startup directly look at the feedback from the system that contains these velocities, see Figure 7. Without doing so our internal velocity prediction step will assume that the velocity is greater than the robot platform can deliver. When the vehicle has started to drive we then can utilize our acceleration models used within the optimization to predict more velocities.

As the MPC scheme to a great extent solves very similar problems from one iteration to the next, we also utilize a “warm-start”, which is to provide the previous solution to the optimization problem as an initial start configuration. Performing the navigation task though the doorway as depicted in Figure 16 with warm start had an average optimization time of 29±3 ms compared to 34±5 ms without warm start, a reduction of approximately 10–20%.

### 5.5. Perception

As mentioned in previous sections, the amount of constraints per added obstacles can be rather high and depends on the footprint representation and the horizon length *N*. To keep the amount of obstacle points o low we have to do some “clever” filtering of the range data. An example on filtered data is depicted in Figure 15. At a higher level, we have a perception system that is based on a previously built occupancy map which is overlaid with LiDAR data. This occupancy map is used to generate the global path. As we use this global path to extract the local path we do have some obstacle-avoidance systems in place. The global path planner has two main parameters that are of importance. First, to simplify the global motion planning the platform’s footprint is approximated with a circle, to allow the robot to traverse narrow areas. The circle does not enclose the full actual footprint but ensures that the vehicle can pass given that its orientation is suitable. The second parameter is the distance from any object or occupied cell that will cause the planned motion to have additional cost. Given an empty environment and a wall, this distance will ensure that the generated plan will always keep this distance away from the wall, unless the goal is given closer towards the wall. These two parameters ensure that the global plan finds traversable areas in constrained areas while it makes sure that the the robot keep some distance in areas where there is space.

From the LiDAR we get a set of range readings that can be transformed into a corresponding 3D coordinate depending on sensor placement and internal parameters. Around the yaw rotation in the robot frame, a sector is defined where the closest reading that is considered to be an obstacle is stored (for example, readings that are sampled from the floor or objects that are higher than the actual height of the robot are ignored). To simplify processing the order, the sector is sorted based on the corresponding yaw angle. From this set of sectors we would like to extract the *O* closest obstacles. From all sectors we choose the closest reading as the first obstacle. It would be likely that the next closest obstacle is from a sector close by, but instead of looking solely at the second closest we make sure that the distance between the selected readings is at least a threshold distance apart. By doing so we can drastically reduce the amount of obstacles while still providing a set that can reasonably well represent the spatial layout of the environment. These obstacle readings will be continuously updated at the next iteration so obstacles that were filtered away might now be visible. Another aspect is that the global plan provided will keep a path with a distance to obstacles. The obstacles here will be used as constraint in the optimization framework, meaning that they will only impact the control actions taken if the obstacles are within the specified collision bounds. This requires that the obstacle is sufficiently close to the robot.

## 6. Experiments and Analysis

The MSDU Local Planner was evaluated in a number of real-world experiments on the robotic platform. The evaluation was performed in a research lab, and initially the robot was manually driven through the environment to build a global map using the LiDAR sensors (see Figure 8 for the created map). This map was used for planning and localization. The map was not updated during the experiments, which took place over several weeks, even though minor modifications occurred within the environment, with obstacles appearing and disappearing. Nonetheless, the system was able to successfully localize and plan.

The experiments performed evaluated the efficiency and accuracy of the paths planned by the MSDU Local Planner, and compared its performance to the TEB Local Planner [9], a widely used local planner. Further experiments evaluated the performance of MSDU in specific scenarios relating to particularly challenging environment aspects, such as narrow gaps and doorways.

The first evaluation compared the paths generated by MSDU against TEB. The goal positions were randomly generated in the environment, and the robot drove between the goals positions using MSDU. The experiment was repeated using TEB.

The termination criteria used are the distance between the current pose and goal as well as the maximum allowed velocity. If both these criteria are met the robot is considered to have reached the goal. In order to give as fair comparison as possible, these parameters were tweaked so that for both planners all goals were reachable within reasonable time and without causing excessive turning in the end. It should also be noted that the TEB Local Planner is not targeting pseudo-omnidirectional platform specifically and the parameters used were empirically found to produce outputs that the platform at hand worked well with. This comparison should therefore primarily be seen as a base-line of how an off-the-shelf planner can perform.

The generated paths were evaluated according to the following criteria:Accuracy: how closely did the final robot pose match the desired goal pose? To evaluate the accuracy, the displacement (both translational and angular) between the final robot pose and the desired goal pose was measured for each goal.Efficiency: how far did the robot need to travel to achieve the desired goal pose? To evaluate this measure, the distance traveled between the robot’s start pose and final pose was measured for each goal. Both translational and angular distance were measured. It was decided that distance traveled was a fairer measure of efficiency for the path planners rather than time taken, as the time taken was too dependent on external parameters to the planners such as the maximum allowed velocity.

The localization performance was evaluated using the robot’s pre-generated map and pose.

The experiment was performed across both “short” and “long” paths. Short paths are particularly challenging for the local motion planners, as they have to take tighter, more confined routes to the goal position. Here, “short” paths had a direct distance between starting and ending positions of less than 1 m. A total of 69 short paths were tested, with a distance normally distributed around a mean of 0.55 m, with a maximum distance of 0.98 m and a minimum distance of 0.12 m. The longer, less challenging paths had a mean distance between starting and ending positions of 2.9 m, with a maximum distance of 3.8 m and a minimum distance of 2.3 m. The experiments are summarized in Table 1.

Repeatability experiments were also performed, where the robot traveled repeatedly between two goals. The first repeatability experiment was over a long path between two goals 2.4 m apart, while the second repeatability experiment was over a short path between two goals 0.42 m apart. The robot traveled to each goal position 20 times. The repeatability experiments are summarized in Table 2.

## 7. Results

This section presents the experimental evaluation of the MSDU Local Planner. Section 7.1 presents the quantitative experiments evaluating the paths generated by MSDU and TEB. Section 7.2 and Section 7.3 present qualitative descriptions of the performance of MSDU in narrow and confined spaces. Finally, Section 7.4 provides a brief summary of the main conclusions from the path evaluation experiments.

### 7.1. Accuracy and Efficiency of Planned Paths

Figure 9 presents the error (both translational and angular) between the final robot pose and the desired goal pose for each goal, for both MSDU and TEB for “long” (2.3–3.8 m) paths. Some key metrics are also summarized in Table 3. It can be seen (Figure 9a) that both planners achieved a high accuracy in terms of linear position. However, MSDU typically has a smaller translational error than TEB: MSDU has a mean error of 0.0019 m from the goal while TEB has a mean error of 0.0031 m from the goal position.

Similar results can be seen when it comes to angular error (see Figure 9b and Table 3). MSDU has a mean angular error of 0.0007 rad while TEB has a mean angular error of 0.0025 rad.

Figure 10 and Table 4 present the distance traveled by the robot to the final goal position. It can be seen that TEB consistently takes longer paths to the goal than MSDU. This is particularly the case for angular distance, where in the worst case TEB travels 20 times the angular distance of MSDU (Goal 18).

The results for the short paths can be seen in Figure 11 and Table 5. The first thing to note is that the short paths are more challenging for both planners than the long paths: the mean (translational, angular) error increases from (0.0019 m, 0.0007 rad) on the long paths to (0.0028 m, 0.0010 rad) on the short paths for MSDU, and from (0.0031 m, 0.0025 rad) on the long paths to (0.0033 m, 0.0038 rad) on the short paths for TEB. However, MSDU continues to outperform TEB in terms of accuracy.

For the short paths, the distance traveled to each goal pose is displayed in Figure 12 and Table 6. The translational distance traveled is longer for the TEB in 97% of the experiments (67 out of 69). In the cases where the translational distance traveled by TEB is shorter, the MSDU path is no more than 1.07 times as long as the TEB path (0.51 m to 0.48 m), while when the TEB path is longer; in the worst case it is 1.78 times as long as MSDU (0.98 m to 0.55 m).

As with the long paths, the angular distance traveled by TEB is consistently longer than that traveled by MSDU: in the worst case TEB takes an angular distance 51 times as long as that of MSDU (Goal 47).

These results show that the MSDU Local Planner can achieve final robot poses that are both closer to the desired goal pose while traveling shorter distances both in terms of translational and angular displacement.

To understand why the distances traveled differ so much between planners, we display a number of the trajectories taken by the robot. We display the trajectories for the first four short goals in Figure 13. In each figure, the goal is marked by the black circle, the trajectory planned by MSDU is displayed in black and the trajectory planned by TEB is displayed in red. It is important to note that the trajectories, even when generally smooth, appear to have a slightly “jagged” appearance. This does not represent the actual path of the robot, but is due to the localization process, where the dead-reckoning estimate based on odometry is updated periodically by the AMCL localization process, thus creating small apparent “jumps” in the path.

However, it is still possible to see the deviation between the paths planned by MSDU and those planned by TEB. TEB often takes a wider, more sweeping path—as one might see from a more car-like vehicle—and sometimes (as in the case of Goal 2) must do some complicated corrections close to the goal. In comparison, MSDU takes a more direct path, and very little correction occurs close to the goal.

The repeatability of the planners is displayed in Figure 14 below, for both the long trajectory (Figure 14a) and the short trajectory (Figure 14b). The trajectory of the robot using MSDU is displayed in black, and the trajectory of the robot using TEB is displayed in red. The qualitative difference between the two planners’ behavior can be clearly seen. The quantitative comparison is presented in Table 7 and Table 8. A key distinction between the two planners is that in each case the distance MSDU travels to Goal 1 is very similar to the distance it travels to Goal 2, while TEB takes quite different-length paths between the two goals for the short paths. This can also be observed in Figure 14b.

### 7.2. Local Planning Capabilities

To give an intuitive understanding of what types of problem the local planner can address, and to also give some illustration regarding how obstacles are extracted and how collision constraints are formed, we performed the following task. When the robot was located in a doorway, we performed a $180^{\circ }$ turn on the spot. As the global planner does not have a notion of the shape of the robot, it just gave the goal location directly. However, due to the shape of the vehicle it could not turn on the spot without causing a collision between the robot and the door frame; instead it had to move out of the doorway and there perform the rotation before entering the doorway again. This is a problem that the optimization-based local planner addressed out of the box and snapshots of this turn are depicted in Figure 15.

### 7.3. Passing a Doorway

To illustrate the connection between the global planner and the proposed local planner when passing through a doorway, a sequence of images are depicted in Figure 16. The local goal orientation is computed by the directions given by the consecutive points before and after the current local goal point.

### 7.4. Summary of Results

The experiments demonstrate that MSDU consistently generated paths for the vehicle that were more accurate than those planned by TEB. Over the long paths in the experimental evaluation, MSDU had a mean translational error of 0.0019 m while TEB had a mean translational error of 0.0031 m. The mean angular error for MSDU was 0.0007 rad compared to 0.0025 rad for TEB. Over the (more challenging) short paths, MSDU had a mean translational error of 0.0028 m while TEB had a mean translational error of 0.0033 m. The mean angular error for MSDU was 0.0010 rad compared to 0.0038 rad for TEB.

The MSDU paths were also shorter than the TEB paths. Over the long paths, MSDU had a mean translational distance of 3.0011 m while TEB had a mean translational distance of 3.1877 m. Over the short paths, MSDU had a mean translational distance of 0.6026 m while TEB had a mean translational distance of 0.7346 m. In terms of angular distance traveled, MSDU had a mean angular distance traveled of 1.1690 rad over the long paths and 1.6130 rad over the short paths, while TEB had a mean angular distance traveled of 3.8198 rad over the long paths and 3.7598 rad over the short paths. In the worst-case scenario the TEB path traversed an angular distance 51 times as long as that of MSDU.

## 8. Conclusions

This work presents a local planner for mobile autonomous platforms with combined steer and drive wheels. It uses a non-linear optimization formulation to handle the kinematic characteristics of the wheel configuration, the physical limitations of the platform, and the obstacles in the environment, to generate a trajectory that generates smooth and efficient paths to a goal, and achieves highly accurate goal pose positions.

The local planner is tightly integrated into a full localization, navigation and motion-planning pipeline, and utilizes a global planner that is continuously re-executed to give an up-to-date global feasible path with respect to the latest readings and map updates. This enables the local planner to look exclusively at local problems and provide an online control signal based on immediate sensor and motion data, while the global planner resolves problems that require a broader global knowledge of the environment.

An interesting future research direction is how dynamic obstacles in the environment—such as people, or other autonomous robots—could be managed by the system. An interesting and non-trivial question to ask is in which layer of the system the management of dynamic obstacles should be included. For example, if we obtain a prior knowledge of typical movement in the environment then this would be suitable to use when providing the global plan. Dynamic obstacles can also be relatively straightforwardly formulated as constraints connected to the predicted horizon if we have a model of how these obstacles move in the future.

In the current formulation, the actual steering wheel direction is not explicitly modeled. It would be of interest to study how to incorporate these in an efficient way. If we can look at the steering angle and the rate of change directly as constraint many detours taken in the current approach, such as taking care of the ICR, would indirectly be handled. One problem is that with a set of four steer-and-drive wheels there is a significant amount of redundancy that is far from straightforward to put together as an optimization problem that both gives reasonable outputs and is also sufficiently fast to solve.

## Figures and Tables

**Figure 1 sensors-22-02588-f001:**
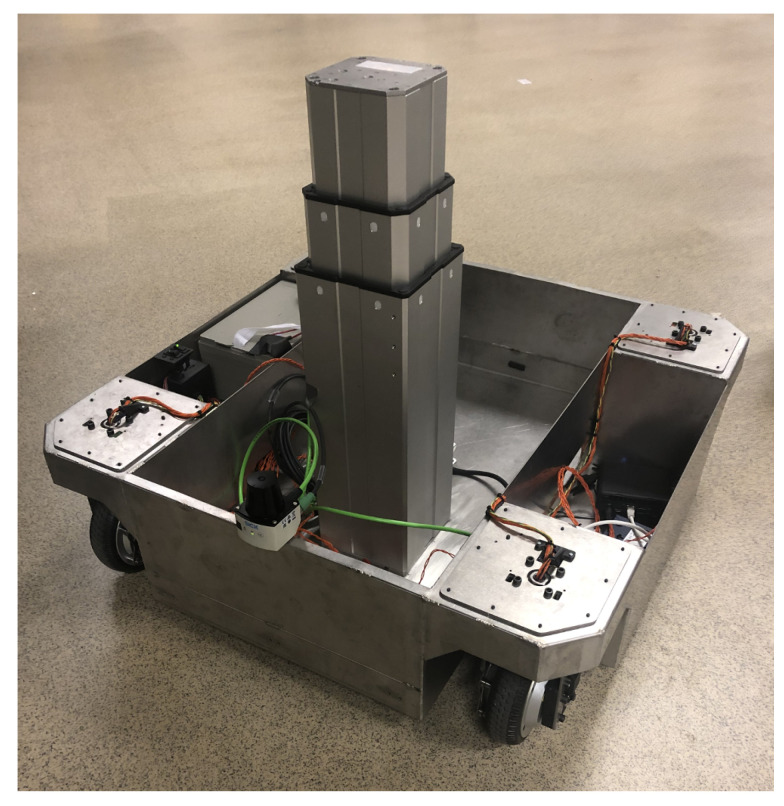
Robotic platform used in the evaluation. It has four combined steer and drive wheels located at each corner. The onboard computer is an Intel NUC i7.

**Figure 2 sensors-22-02588-f002:**
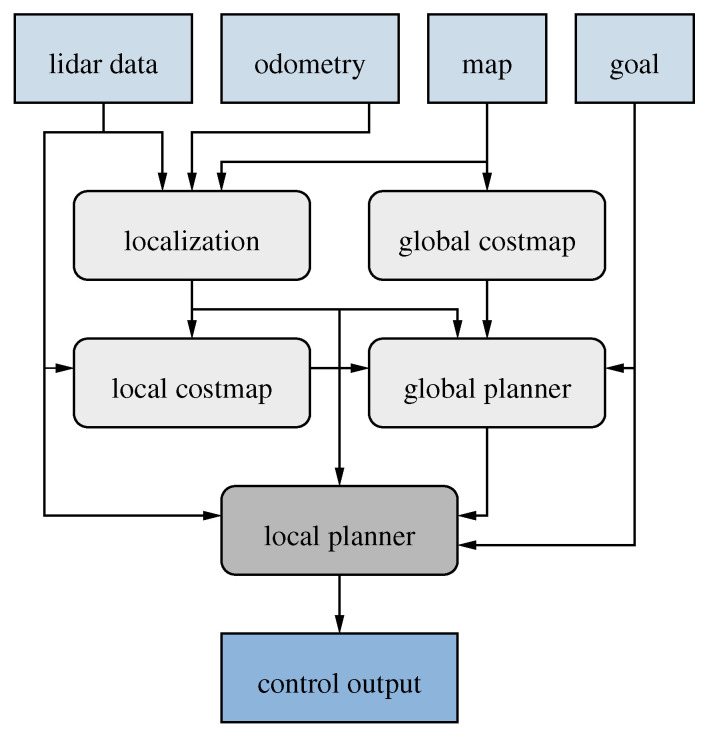
Overview of the different components of the localization, navigation and planning system and their interactions.

**Figure 3 sensors-22-02588-f003:**
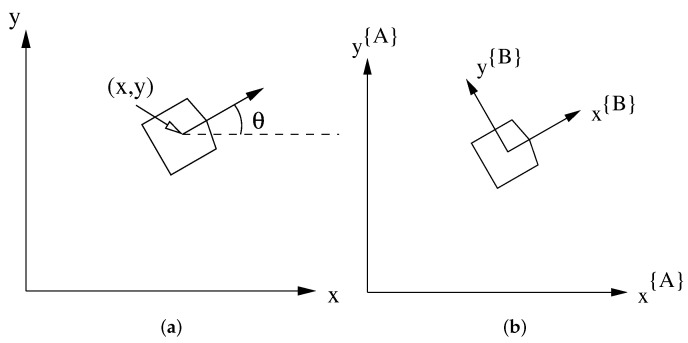
Platform pose and coordinate frames used. (**a**) The platform pose is denoted by position (x,y) and orientation θ in a world frame; (**b**) the relationship between the world coordinate frame {A} and the platform coordinate frame {B}.

**Figure 4 sensors-22-02588-f004:**
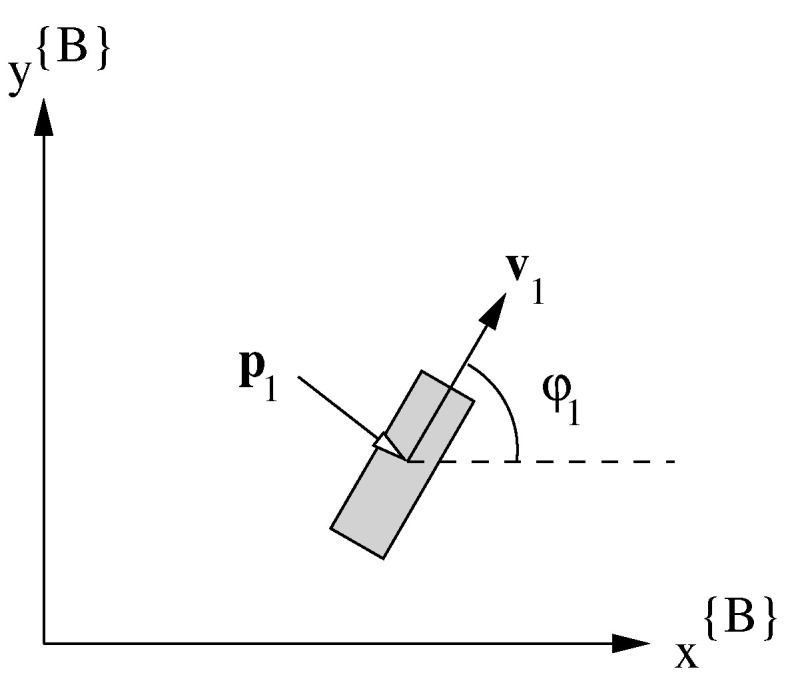
Wheel parameters: $p_{1}$ indicates the position, $v_{1}$ the velocity vector, $v_{i}=∥v_{i}∥$ the velocity (scalar) and $\varphi_{1}$ is the steering (scalar).

**Figure 5 sensors-22-02588-f005:**
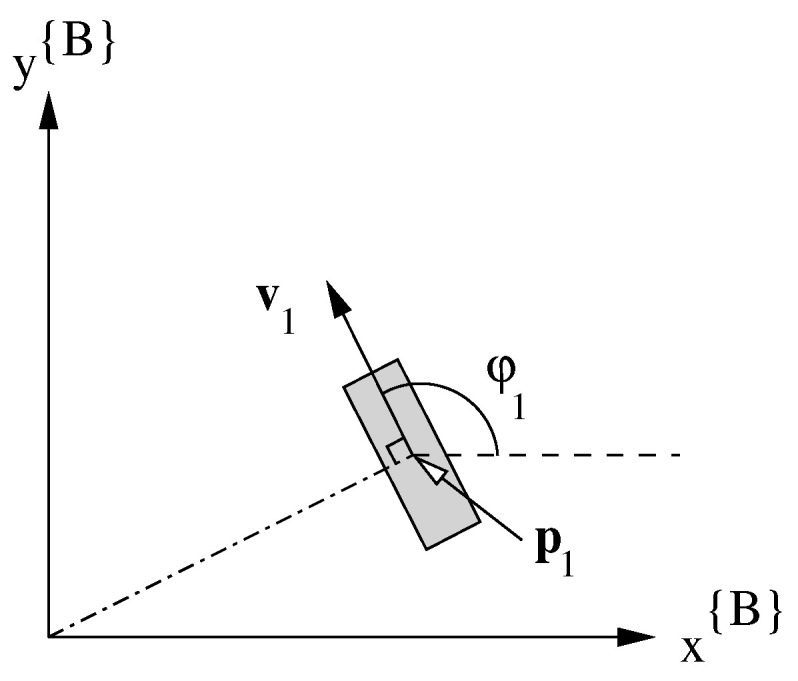
Wheel 1 parameters given that v=0, which enforces the steering wheel angle $\varphi_{1}$ to make the steering direction vector $v_{1}$ to be orthogonal towards the origin of the frame.

**Figure 6 sensors-22-02588-f006:**
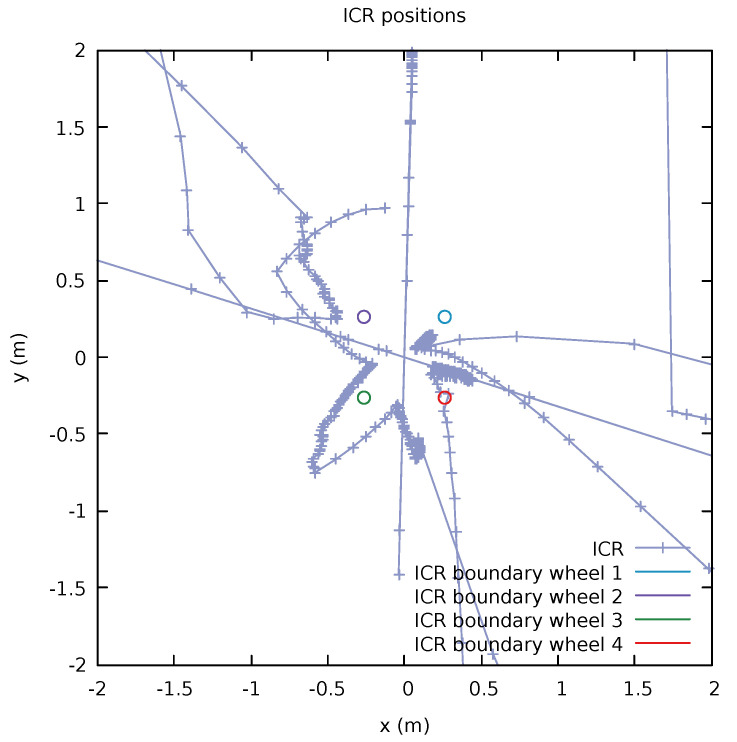
An example of the locations of ICR when moving around the platform, where the line represents the the consecutive changes of ICR locations. One can see the effect of ω (Equation (Equation 7)) that a relatively large change in ICR happens when there is low rotational velocity ω. In this example, we can see that the ICR was close to the boundary of wheel 4 (red) and probably was adjusted to stay outside the boundary. Note that to smoothly move from pure rotational velocity to linear velocity the ICR point has to incrementally move from the center passing in between the ICR constraint regions, therefore it is important to make the ICR constraint region not larger than needed.

**Figure 7 sensors-22-02588-f007:**
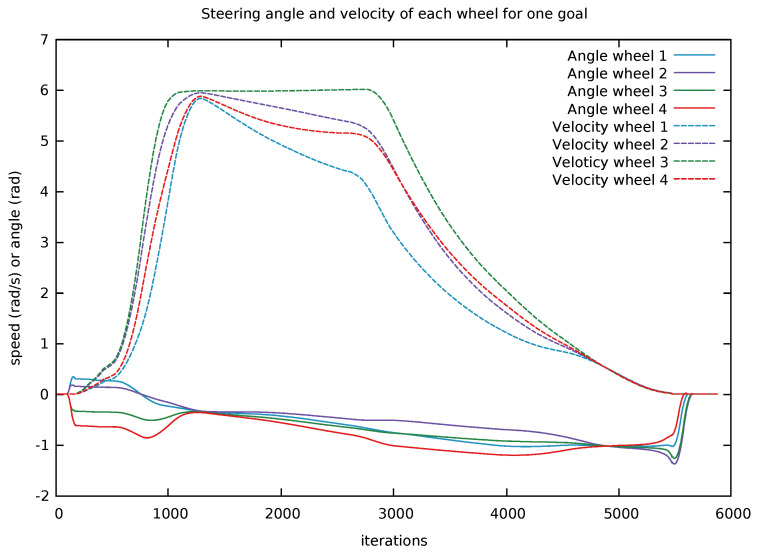
An example output driving from a start pose to a goal pose showing the heading and velocity of all four steer and drive units. In the beginning, the steering wheels all have an angular heading of zero. To allow the vehicle to start driving at an arbitrary direction, we utilize the feedback given by the platform to ensure that all wheels are aligned before the velocity is applied. When the vehicle has reached its goal the heading angles is set back to zero by the internal controller.

**Figure 8 sensors-22-02588-f008:**
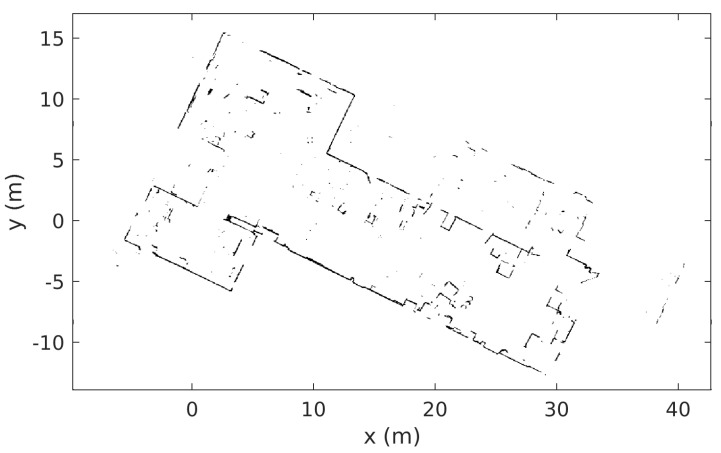
The map generated by the robot and used for localization, navigation, and planning. The robot was manually driven around the environment to create the map before the experiments took place.

**Figure 9 sensors-22-02588-f009:**
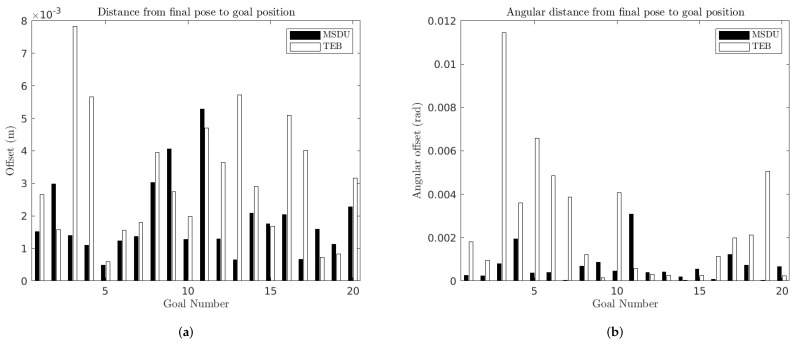
Distance between final robot pose and desired goal position for the MSDU Local Planner (black) and the TEB Local Planner (white) over long paths. (**a**) Translational error between the final robot pose and the desired goal position; (**b**) angular error between the final robot pose and the desired goal position.

**Figure 10 sensors-22-02588-f010:**
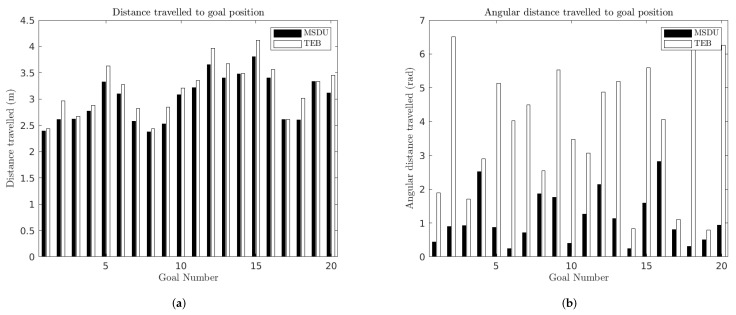
Distance traveled between the starting and final robot pose for the MSDU Local Planner (black) and the TEB Local Planner (white) over long paths. (**a**) Translational distance traveled between the starting and final robot pose (**b**) Angular distance traveled between the starting and final robot pose.

**Figure 11 sensors-22-02588-f011:**
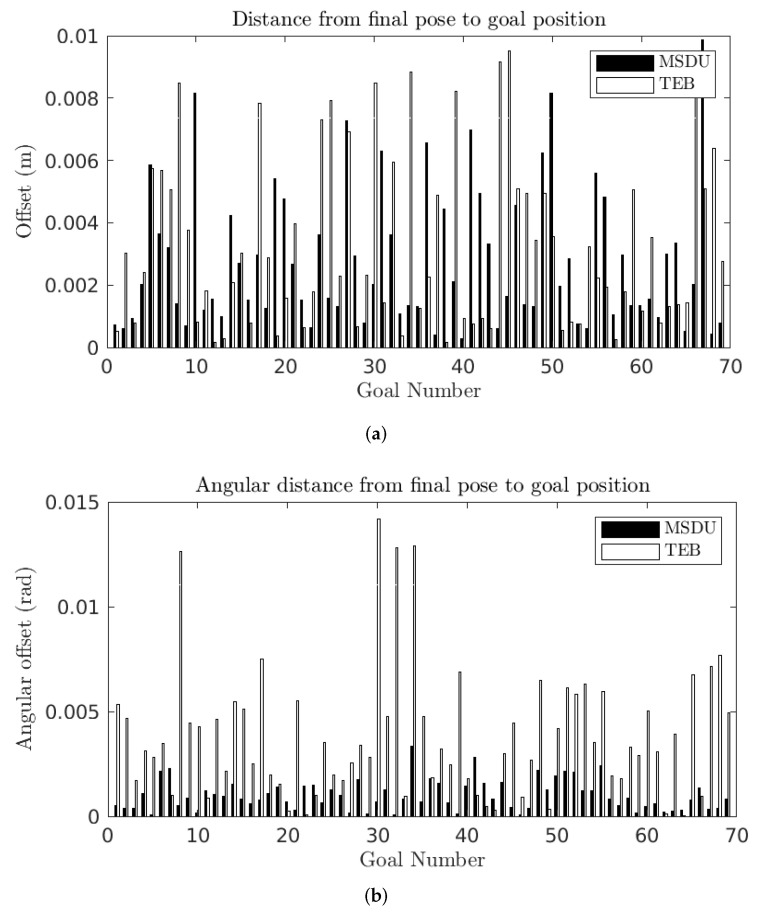
Distance between final robot pose and desired goal position for the MSDU Local Planner (black) and the TEB Local Planner (white) over short (<1 m) paths. (**a**) Translational error between the final robot pose and the desired goal position; (**b**) angular error between the final robot pose and the desired goal position.

**Figure 12 sensors-22-02588-f012:**
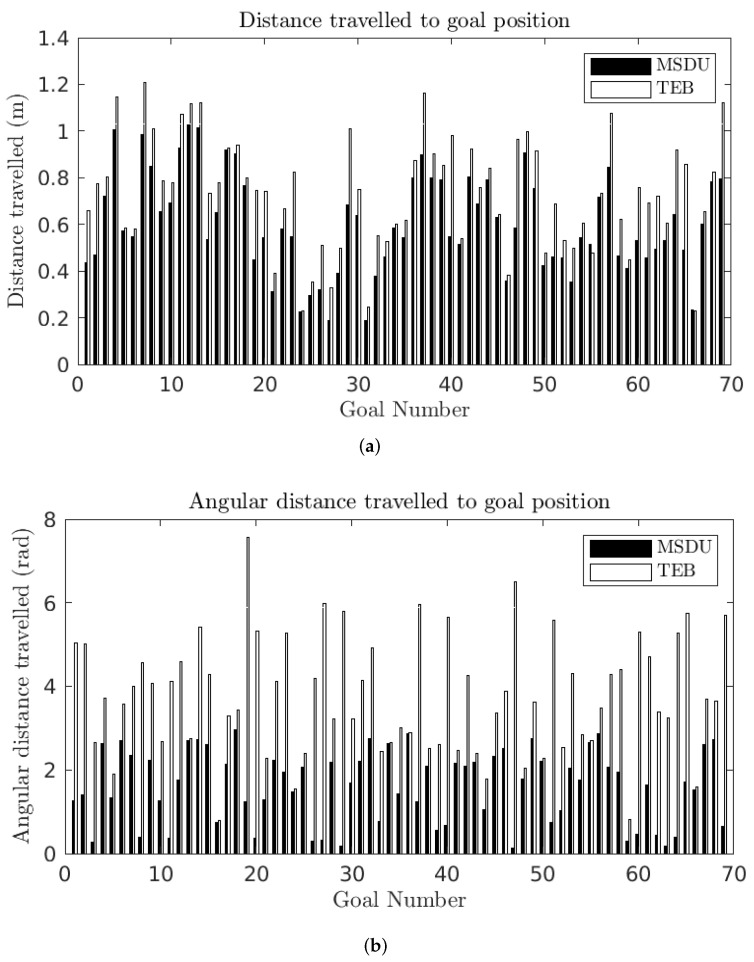
Distance traveled between the starting and final robot pose for the MSDU Local Planner (black) and the TEB Local Planner (white) over short (<1 m) paths. (**a**) Translational distance traveled between the starting and final robot pose; (**b**) angular distance traveled between the starting and final robot pose.

**Figure 13 sensors-22-02588-f013:**
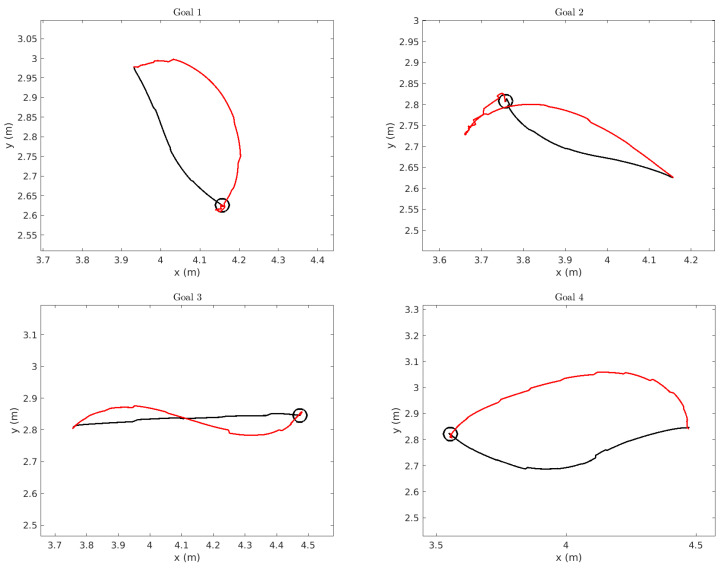
Trajectories taken by the robot for each goal. The goal is denoted by the black circle. The position of the robot using the MSDU Local Planner is displayed in black, and the robot position by the TEB Local Planner is displayed in red. Note that discrepancy in the curves are due to the localization update.

**Figure 14 sensors-22-02588-f014:**
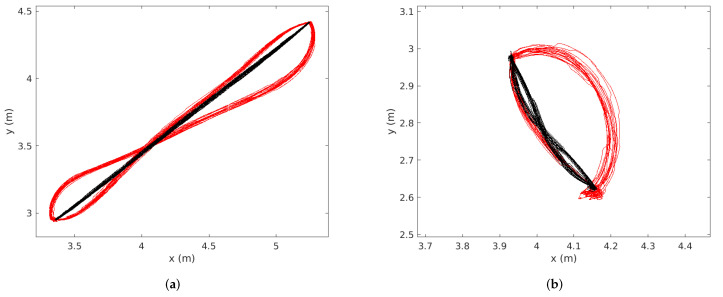
Trajectories taken by the robot during the repeatability experiment. The trajectory of the robot using MSDU is displayed in black, and the robot trajectory using TEB is displayed in red. (**a**) Long path (2.4 m between the two goals); (**b**) short path (0.42 m between the two goals).

**Figure 15 sensors-22-02588-f015:**
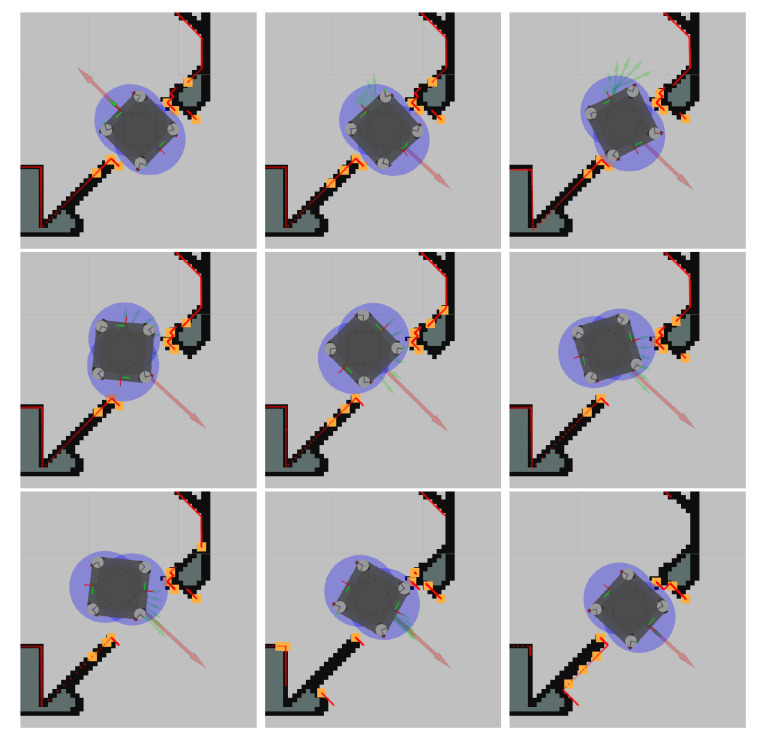
Set of snapshots showing the capability of the local planner to do a $180^{\circ }$ turn inside a narrow doorway where there is not enough space to turn on the spot. Top left is the start configuration and bottom right is the goal configuration. In each picture, the red arrow depicts the current goal. The green arrows depict the future horizon states ${\left(x,y,\theta \right)}_{1\ldots N}$. The blue circles depicts the collision circles $c_{1\ldots C}$. The yellow squares is the extracted obstacle points $o_{1\ldots O}$. The red smaller dots are the LiDAR points.

**Figure 16 sensors-22-02588-f016:**
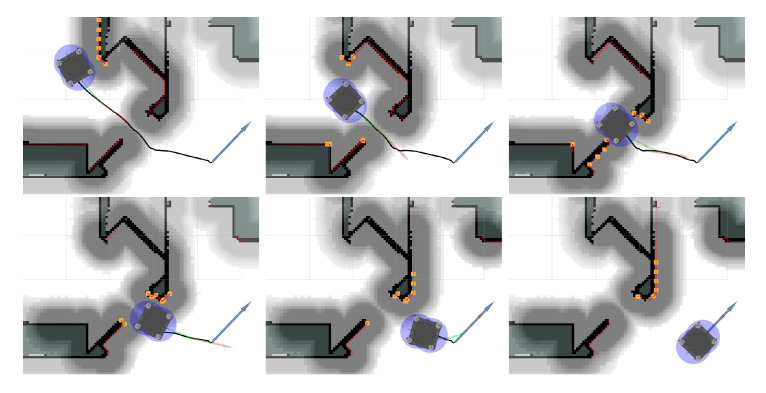
Set of snapshots illustrating the interaction between the global planner (the planned path is depicted as a black line and the goal is light-blue). The red arrow depicts the current local goal that the proposed local planner is driving towards, the green arrows depict the future horizon states (also see the caption in Figure 15. Due to the look ahead, the optimization allows for a smoother driven path compared to the relatively non-smooth and jerky path provided by the global path planner. The additional spacing between the obstacle points and the robot when there is enough space can be seen in the top left figure as the cost utilized by the global planner (illustrated with different gray levels) contains an additional offset. By setting this offset to at least corresponding to a single collision radius of the platform, that the platform can be in any orientation makes the obstacle constraints only to be active in areas where there is simply not enough space. Additionally, this additionally safety margin is very useful as the robot would otherwise drive very close to obstacles as the obstacle constraints do not have any other cost associated with them based on distance.

**Table 1 sensors-22-02588-t001:** Summary of accuracy and precision experiments.

Accuracy and Precision Experiments	Number of Repetitions	Mean Distance (m)	Max Distance (m)	Min Distance (m)
Long paths	20	2.9 m	2.3 m	3.8 m
Short paths	69	0.55 m	0.12 m	0.98 m

**Table 2 sensors-22-02588-t002:** Summary of repeatability experiments.

Repeatability Experiments	Number of Repetitions	Distance between Goals (m)
Long paths	20 per goal (40 total)	2.4 m
Short paths	20 per goal (40 total)	0.42 m

**Table 3 sensors-22-02588-t003:** Summary of performance of MSDU and TEB for long paths.

Translational Error (m)	MSDU	TEB
Mean (m)	0.0019	0.0031
Standard deviation (m)	0.0012	0.0019
Max (m)	0.0053	0.0078
Min (m)	0.0005	0.0006
**Angular error (rad)**		
Mean (rad)	0.0007	0.0025
Standard deviation (rad)	0.0007	0.0029
Max (rad)	0.0031	0.0114
Min (rad)	$4\times 10^{-5}$	$4\times 10^{-5}$

**Table 4 sensors-22-02588-t004:** Summary of distances traveled by MSDU and TEB for long paths.

Translational Distance Traveled (m)	MSDU	TEB
Mean (m)	3.0011	3.1877
Standard deviation (m)	0.4437	0.4823
Max (m)	3.8077	4.1208
Min (m)	2.3772	2.4355
**Angular distance traveled (rad)**		
Mean (rad)	1.1690	3.8198
Standard deviation (rad)	0.7646	1.8938
Max (rad)	2.1898	6.5075
Min (rad)	0.2373	0.7898

**Table 5 sensors-22-02588-t005:** Summary of performance of MSDU and TEB for short paths (<1 m).

Translational Error (m)	MSDU	TEB
Mean (m)	0.0028	0.0033
Standard deviation (m)	0.0022	0.0028
Max (m)	0.0099	0.0095
Min (m)	0.0003	0.0002
**Angular error (rad)**		
Mean (rad)	0.0010	0.0038
Standard deviation (rad)	0.0007	0.0031
Max (rad)	0.0033	0.0142
Min (rad)	$5\times 10^{-5}$	$1\times 10^{-5}$

**Table 6 sensors-22-02588-t006:** Summary of distances traveled by MSDU and TEB for short paths.

Translational Distance Traveled (m)	MSDU	TEB
Mean (m)	0.6026	0.7346
Standard deviation (m)	0.2114	0.2390
Max (m)	1.0260	1.2085
Min (m)	0.1869	0.2822
**Angular distance traveled (rad)**		
Mean (rad)	1.6130	3.7598
Standard deviation (rad)	0.8756	1.4064
Max (rad)	2.9687	7.5681
Min (rad)	0.1260	0.7815

**Table 7 sensors-22-02588-t007:** Repeatability of performance of MSDU and TEB on long paths.

Goal Pose Error	MSDU (Mean, Std)	TEB (Mean, Std)
Goal 1 translational error (m)	(0.0021, 0.0014)	(0.0057, 0.0027)
Goal 2 translational error (m)	(0.0027, 0.0015)	(0.0030, 0.0014)
Goal 1 angular error (rad)	($5\times 10^{-4}$, $6\times 10^{-4}$)	(0.0040, 0.0025)
Goal 2 angular error (rad)	($6\times 10^{-4}$, $4\times 10^{-4}$)	(0.0034, 0.0030)
**Distance Traveled**		
Goal 1 translational (m)	(2.3971, 0.0039)	(2.4384, 0.0072)
Goal 2 translational (m)	(2.3960, 0.0047)	(2.7068, 0.0213)
Goal 1 angular (rad)	(0.4414, 0.0038)	(1.9345, 0.0452)
Goal 2 angular (rad)	(0.4398, 0.0050)	(5.4977, 0.0399)

**Table 8 sensors-22-02588-t008:** Repeatability of performance of MSDU and TEB on short paths.

Goal Pose Error	MSDU (Mean, Std)	TEB (Mean, Std)
Goal 1 translational error (m)	(0.0025, 0.0018)	(0.0029, 0.0014)
Goal 2 translational error (m)	(0.0033, 0.0018)	(0.0028, 0.0017)
Goal 1 angular error (rad)	($6\times 10^{-4}$, $4\times 10^{-4}$)	(0.0035, 0.0022)
Goal 2 angular error (rad)	($7\times 10^{-4}$, $5\times 10^{-4}$)	(0.0051, 0.0042)
**Distance Traveled**		
Goal 1 translational (m)	(0.4418, 0.0087)	(0.6915, 0.0241)
Goal 2 translational (m)	(0.4467, 0.0087)	(0.4581, 0.0079)
Goal 1 angular (rad)	(1.2668, 0.0016)	(5.0739, 0.0161)
Goal 2 angular (rad)	(1.2663, 0.0013)	(1.3190, 0.0212)

## Data Availability

Not applicable.

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
