# Peer review of "A Local Planner for Accurate Positioning for a Multiple Steer-and-Drive Unit Vehicle Using Non-Linear Optimization"

_sensors, 2022, doi:10.3390/s22072588_

Round 1
Reviewer 1 Report
This manuscript is quite interesting. It can be enhanced by including more details from the abstract to the conclusion sections. For example, in the sentences in the abstract you may discuss the following issues:
Abstract: This paper presents a local planning approach that is targeted for pseudo-omnidirectional 1
vehicles: that is, vehicles that can drive sideways and rotate on the spot but which are prone to 2
singular configurations. … you may include some examples of sigular configurations in a form like this: singular configurations, namely:… The presented approach – MSDU Local Planner – is based on optimal 3
control and formulates a non-linear optimization problem formulation that exploits the omni-motion 4
capabilities of the vehicle to drive the vehicle to the goal in a smooth and efficient manner while 5
avoiding obstacles and singularities…. In this sentence, you are encouraged to talk in present tense, since you are going to introduce your main findings and for past tense you can write this style in the conclusions, when you are detailing waht you did. So that the Word “presented” must be changed to “present” to be updated as: The present approach – MSDU-…
The presented approach has been developed for a real platform 6
for mobile manipulation where one key functionality is the capability to drive in narrow and confined 7
areas…. This sentence is similar to the previous one and must be changed to avoid repetition of words, you magy consider writting for exampel: Our proposed approach is developed for a real …. You can see that also the Word “developed” must be changed to present tense and so on… finally, the last sentence in your abstract: The real-world evaluations shows that the generated motions are smooth and provides good 8 position accuracy…. If you can, add quantities with respect to similar Works or similar to the state of the art to better appreciate your main contribution.
You are encouraged to add references in the introduction to appreciate recent and related Works, for instance, the first paragraph in the first section sounds good but there is not support incluidng references, you may include some ones in the brackets recommended as follows:
- Introduction 11
In industrial applications where robot productivity and agility is important [ ], omni- 12
motion capability – the ability to drive in all directions [ ], not just forwards – is clearly 13
advantageous on a mobile industrial robot [ ]. Omni-motion capability allows the robot 14
greater maneuverability [ ], so it can avoid complex maneuvers, which can be difficult to 15
plan [ ], not predictable for people and other robots in proximity [ ], time-consuming [ ], and even 16
impossible in crowded or constrained physical environments [ ]. Furthermore, greater mobile 17
maneuverability allows a robot to achieve more accurate pose targets [ ], even when interacting 18
with other machines, and can be essential for many industrial applications such as mobile 19
manipulation [ ].
In fact, in the Introduction section 1, you only have 4 references. This is not good to show the state of the art, so taht you must include a more detailed motivation including recent and related Works to appreaciate what is the main problema and how are you planning to improve the related challenges.
Sectio 2 related work, must be merged with section 1, or you can include Tables in section 2 to appreciate the main challenges and the related Works.
Section 3 is good but it needs a summary of the guidelines and the challenges that you are triying to solve with optimization. So that section 5. Defining the Optimization Problem can be better appreciated in a subsection called problema formulation.
Section 7 Results is quite long and it can be shortened with some material moved to a new section labeled “Discussion”, so that section 8. Discussion and Conclusions must also be divided and the conclusions can be in a single section 9.
Reviewer 2 Report
The paper presents a local planning approach for vehicles with some singular configurations using a nonlinear programming approach. At first, the authors identify possible solution for providing omnidirectional driving capability, referring it to the literature. This gives rise to the need to create a motion planner to cope with singular configurations in robot's position, impeding the procedure of obtaining the inverse kinematic soltution.
Secondly, the authors present the state of the art solutions from the field, mentioning DWA or TEB ones, leading to the need to obtain solutions in an occluded environment, allowing collision-free movement. Finally they mention a member of a large family, namely RRT algorithm, but do not mention its versions, such as A star, blossom, or fnd for dynamic environments.
I have found neither any information about the contribution, nor about the novelty at the first part of the paper, what is strange.
It is actually expression (8) where any information about time is used, i.e. the reader is informed that it is an ODE, thus the authors use a continuous-time formulation - this should be given much earlier!
Any problems with tan (5) function, and singular configurations? Shouldn't atan2 be used to give the information about the angle?
At the end of Section 4 there is no slip modelled, thus the authors assume there is no greater impact of the surface on the wheel movement.
Line 333 seems ackward. The constraints are not formulated, but they usually result from the environmental condition, natural construction constraints, etc. What if there is no feasible solution for a given reference and a set of constraints imposed? Do you use any strategy to turn hard constraints into soft constraints or any penalizing terms?
The major part of the paper is missing, as the authors actually did not present any MSDU planner at all in a concise form. Where are the constraints, admissible ranges, cost function discussion, simplified models of the robot, etc? Instead we have a textual output presenting some narration, but with hardly any mathematics.
How many repetitions were done? What was the noise level from the sensors? Can you estimate the uncertainty budget from your approach?
The authors should deeply revise the paper, giving all information in a concise way.
Reviewer 3 Report
In the paper, the authors presented a local planning approach that is targeted for pseudo-omnidirectional vehicles: that is, vehicles that can drive sideways and rotate on the spot but which are prone to singular configurations. The presented approach – MSDU Local Planner – is based on optimal control and formulates a non-linear optimization problem formulation that exploits the omni-motion capabilities of the vehicle to drive the vehicle to the goal in a smooth and efficient manner while avoiding obstacles and singularities. The presented approach has been developed for a real platform for mobile manipulation where one key functionality is the capability to drive in narrow and confined areas. The real-world evaluations shows that the generated motions are smooth and provides good position accuracy. Generally, this is a quite interesting work. It can be accepted if the authors can consider the following issues:
- It seems that this is application-oriented work. Then, what is the academic contribution for a academic journal? The authors should well organize the original contribution.
- For the optimization problem with constraint, how did the authors obtain the results? Is it easy for the implementation?
- It seems that this is a over-actuator tracking problem. More related works are welcome to enrich the literature review such as Path Following Control of Autonomous Four-Wheel-Independent-Drive Electric Vehicles Via Second-Order Sliding Mode and Nonlinear Disturbance Observer Techniques;
- More words are welcome for Fig. 6.
- Comparisons are welcome.
Reviewer 4 Report
The authors discuss the problem of motion planner for an autonomous mobile robotic platform. The paper is well structured. A proofread is recommended. Are minor typos that must be corrected (e.g."As the output of a global planner is a typically a path..."). The reference list is proper and up to date. The novelty of the paper must be described in more detail, "a specialized" planner is not enough. The optimization problem must be detailed. How this cost function is solved? How it is implemented? Please offer more details. In present form a reader can not reproduce your results.
Round 2
Reviewer 1 Report
the updated version of this manuscript can be accepted as it is
Reviewer 2 Report
Thank you for taking my comments into consideration. I am fully satisfied with the current form of the paper, and suggest its publication in the current form. Good luck with the review process!